# Performance Improvement of Quantum Dot Light-Emitting Diodes Using a ZnMgO Electron Transport Layer with a Core/Shell Structure

**DOI:** 10.3390/ma16020600

**Published:** 2023-01-08

**Authors:** Ye-Bin Eun, Gyeong-Pil Jang, Ji-Hun Yang, Su-Young Kim, Young-Bin Chae, Mi-Young Ha, Dae-Gyu Moon, Chang-Kyo Kim

**Affiliations:** 1Department of Electronic Materials, Devices and Equipment Engineering, Soonchunhyang University, Asan 31538, Chungnam, Republic of Korea; 2Display New Technology Institute, Soonchunhyang University, Asan 31538, Chungnam, Republic of Korea

**Keywords:** quantum dot light-emitting diode, solution process, ZnMgO nanoparticles, core/shell structure, charge balance

## Abstract

Highly efficient and all-solution processed quantum dot light-emitting diodes (QLEDs) with high performance are demonstrated by employing ZnMgO nanoparticles (NPs) with core/shell structure used as an electron transport layer (ETL). Mg-doping in ZnO NPs exhibits a different electronic structure and degree of electron mobility. A key processing step for synthesizing ZnMgO NPs with core/shell structure is adding Mg in the solution in addition to the remaining Mg and Zn ions after the core formation process. This enhanced Mg content in the shell layer compared with that of the core X-ray photoelectron spectroscopy showed a higher number of oxygen vacancies for the ZnMgO core/shell structure, thereby enhancing the charge balance in the emitting layer and improving device efficiency. The QLED incorporating the as synthesized ZnMgO NP core/shell A exhibited a maximum luminance of 55,137.3 cd/m^2^, maximum current efficiency of 58.0 cd/A and power efficiency of 23.3 lm/W. The maximum current efficiency and power efficiency of the QLED with ZnMgO NP core/shell A improved by as much as 156.3% and 113.8%, respectively, compared to the QLED with a Zn_0.9_Mg_0.1_O NP ETL, thus demonstrating the benefits of ZnMgO NPs with the specified core/shell structure.

## 1. Introduction

Since the first demonstration of a multilayered quantum dot (QD) light-emitting diode (QLED) in 1994 [1], colloidal QLEDs have continued to attract much attention for use in displays and solid-state lighting due to their superior properties, including size-tunable emission colors, saturated color emission, high stability and a solution-based fabrication process [1,2,3,4,5,6,7]. QLED performance has steadily improved in terms of brightness, luminous efficiency and stability due to advances in QD material and charge-transport material, as well as device architecture [8,9,10,11,12,13,14,15,16,17,18,19]. hus, QLED performance is now approaching that of organic LEDs.

In typical QLEDs, the energy barrier for electron injection from the cathode to the emitting layer (EML) is smaller than that of the holes from the anode to the EML. ZnO nanoparticles (NPs) are widely employed in the electron transport layer (ETL) of QLEDs, given that the mobility of electrons in ZnO NPs is much larger than that of holes in the hole transport layer (HTL) [20]. These mismatches result in an electron–hole charge imbalance in the QD EML, thus increasing the possibility of non-radiative exciton recombination and deterioration of device performance [21,22,23]. In addition, a high number of oxygen-deficient surface states in ZnO NPs adsorb molecular O_2_ and H_2_O from the air [20]. The adsorbed oxygen can lead to non-radiative recombination at the interface of the QD EML and ZnO NP ETL, resulting in a reduction in device efficiency.

An approach to enhance the balance of charges in the EML is to suppress the transport of electrons in the ETL [21,24,25]. For example, to reduce the injection of electrons from the ETL to the EML, Peng et al. inserted a 6-nm-thick insulating poly(methyl methacrylate) layer between the QD EML and ZnO NP ETL to create a highly efficient QLED with an external quantum efficiency of 20.5% [12]. Wu et al. demonstrated a perovskite nanocrystal (NC) LED with a ZnMgO interfacial layer; the air stability of the perovskite NC film was significantly enhanced due to a reduction in the oxygen vacancy surface states of ZnMgO NCs [26].

Kim et al. fabricated QLEDs with Mg-doped ZnO NPs (ZnMgO NPs) using different doping concentrations; they demonstrated enhanced performance, with a current efficiency of 56.0 cd/A [27]. Mg doping of the ZnO NPs modifies their electronic structure and electron mobility, such that the energy barrier between the cathode and ZnMgO NP ETL increases. This degrades the electron transport performance of the ZnMgO ETL, resulting in an enhanced charge balance in the QD EML. Li et al. reported improved QLED performance by modulating the electron injection with yttrium (Y)-doped ZnO NPs [28]; the Y-doping concentration in the ZnO NPs generated a larger energy barrier between the cathode and ZnO ETL, thus improving device performance. Kim et al. modified the energy band structure of a Li and Mg co-doped ZnO ETL to create a highly efficient QLED; co-doping of Li and Mg in ZnO NPs increased the band gap and electrical resistivity, resulting in more favorable charge balance in the EML [20]. Hwang et al. modified the charge injection by adjusting the thickness of Zn_0.9_Mg_0.1_O NP ETL to increase the resistivity of the ETL [29]. Liu et al. doped ZnO NPs with small organic molecules to modulate the electron transport properties of the ZnO NPs [30]. Chrzanowski et al. reported enhanced efficiency for a QLED using a sol-gel Zn_1−x_Mg_x_O ETL; the increase in Mg content in Zn_1−x_Mg_x_O effectively increased the conduction band minimum (CBM), thus reducing electron mobility, to create a more favorable charge balance in the QD EML [31]. Heo et al. demonstrated improved electron injection and electron transport behaviors for low turn-on QLEDs using a stepwise ZnMgO/ZnO double ETL [24].

In this study, we introduced ZnMgO NPs with a core/shell structure as the ETL in QLEDs. The shell layers of ZnMgO NPs were grown onto the core using the Zn and Mg ions remaining after use during the core process and additionally adding Mg to the solution. Thus, the Mg content in the shell layer was expected to be higher than that in the core layer. The oxygen vacancy content increased in the ZnMgO NPs with the core/shell structure, as confirmed by X-ray photoelectron spectroscopy (XPS), thus reducing electron transport in the ZnMgO NP ETL with core/shell structure. QLEDs were prepared using the ZnMgO NP ETLs such as ZnO NPs, Zn_0.9_Mg_0.1_O NPs, ZnMgO NPs with core/shell structure via a solution process. The electronic structure of the QLEDs was measured by ultraviolet photoelectron spectroscopy (UPS). According to the results, the QLED with the ZnMgO NP ETL with a core/shell structure exhibited a maximum luminance of 55,137.3 cd/m^2^, maximum current efficiency of 58.0 cd/A and maximum power efficiency of 23.3 lm/W; these results are attributable to a change in charge balance induced in the QD EML by Mg doping of the ZnO NPs and the core/shell configuration.

## 2. Materials and Methods

### 2.1. Syntheses of Materials

ZnO NPs and Zn_0.9_Mg_0.1_O NPs in a colloidal suspension were synthesized using a sol-gel method, as described in our previous study [29]. To synthesize ZnO NPs, 0.3292 g of zinc acetate dihydrate (Zn(CH_3_COO)∙2H_2_O) powder (Sigma-Aldrich, St. Louis, MO, USA), as a precursor material, was dissolved in 15 mL of dimethyl sulfoxide (DMSO). Tetramethylammonium hydroxide (TMAH, 0.421 g; Sigma-Aldrich) was dissolved in 5 mL of EtOH. The TMAH and Zn(CH_3_COO)∙2H_2_O solutions were stirred at room temperature for 24 h to ensure that they were fully dissolved. Ethyl acetate (Sigma-Aldrich) was poured into the synthesized solution to precipitate the ZnMgO NPs. The volume ratio of the ethyl acetate and synthesized solution was 3:1. After 3 h, a white powder had formed and precipitated. ZnO NPs were obtained by centrifuging the solution.

To synthesize Zn_0.9_Mg_0.1_O NPs, 0.2962 g of zinc acetate dihydrate (Zn(CH_3_COO)∙2H_2_O) powder (Sigma-Aldrich) and 0.03292 g of magnesium acetate tetrahydrate (Mg(CH_3_COO)∙4H_2_O) powder (Sigma-Aldrich), as precursor materials, were dissolved in 15 mL of DMSO. TMAH (0.421 g) was dissolved in 5 mL of EtOH. The mixture of the TMAH solution and mixed solution of Zn(CH_3_COO)∙2H_2_O and 2Mg(CH_3_COO)∙4H_2_O) was stirred at room temperature for 24 h to ensure that they were fully dissolved. Ethyl acetate (Sigma-Aldrich) was poured into the synthesized solution to precipitate the Zn_0.9_Mg_0.1_O NPs. The volume ratio of the ethyl acetate and synthesized solution was 3:1. After 3 h, a white powder had formed and precipitated. Zn_0.9_Mg_0.1_O NPs were obtained by centrifuging the solution.

To synthesize the ZnMgO NP solution used as the core of the ZnMgO NP core/shell structure, 0.263 g of zinc acetate dihydrate (Zn(CH_3_COO)∙2H_2_O) powder (Sigma-Aldrich) and 0.033 g of magnesium acetate tetrahydrate (Mg(CH_3_COO)∙4H_2_O) powder (Sigma-Aldrich), as precursor materials, were dissolved in 12 mL of DMSO. TMAH (0.421 g) was dissolved in 5 mL of EtOH. The mixture of the TMAH solution and mixed solution of Zn(CH_3_COO)∙2H_2_O and 2Mg(CH_3_COO)∙4H_2_O) was stirred at room temperature for 5 min to complete the synthesis of the ZnMgO NP core.

To synthesize the shell layers of ZnMgO with a core/shell structure, Mg(CH_3_COO)∙4H_2_O powder (Sigma-Aldrich) in amounts of 0.033 and 0.049 g, as precursor materials, was dissolved in 3 mL of DMSO to create two Mg(CH_3_COO)∙4H_2_O solutions (A and B). Each solution was poured into the as synthesized core solution, followed by stirring at room temperature for 24 h to ensure full dissolution. After 3 h, a white powder had formed and precipitated. After centrifuging the solution, ZnMgO NPs with a core/shell structure were obtained. There was Zn in the solution prepared for synthesizing the shell layer. The ZnMgO NP shell layer was grown onto the core using the remaining Zn and Mg ions formed during core synthesis. As such, we believe that the shell layer had more Mg than the core layer. Hereinafter, the ZnMgO NPs-based ETLs synthesized using 0.033 g of Mg(CH_3_COO)∙4H_2_O powder in the shell is denoted as ZnMgO NP core/shell A and that using 0.049 g of Mg(CH_3_COO)∙4H_2_O as ZnMgO NP core/shell B.

### 2.2. Device Fabrication

QLEDs were fabricated using the following process. An indium tin oxide (ITO) thin film with a sheet resistance of ≤10 Ω/□ on glass substrate was patterned by a photolithography method. The patterned ITO was used as an anode. The ITO-patterned glass substrate was ultrasonically cleaned with acetone, isopropyl alcohol, methanol and deionized water. Then, the ITO-patterned glass substrate was spin-coated with three layers, as described in the following.

Poly(N-vinyl-carbazole) (PVK) (Sigma-Aldrich) was dissolved to 1.2 wt% in toluene; the dissolved PVK solution was deposited onto as-patterned ITO glass substrate by spin-coating at a speed of 600 rpm for 5 s at room temperature, followed by spin-coating at a speed of 1500 rpm for 15 s at room temperature. Then, CdSe/ZnS QDs (Zeus) were dissolved at 5 mg/mL in heptane to create the EML; this solution was then spin-coated onto the ITO/PVK substrate at a speed of 3000 rpm for 5 s at room temperature. Solutions of ZnMgO-based ETLs comprising ZnO NPs, ZnMgO NPs and ZnMgO NPs with a core/shell structure (A/B) were spin-coated onto the ITO/PVK/QD substrate at a speed of 2000 rpm for 20 s at room temperature. Then, the multilayered substrates were loaded into a high-vacuum deposition chamber (Cetus OL 100; Celcose, Hwasung, Korea) (background pressure, 6 × 10^−7^ Torr) to deposit the Al cathode layer (150 nm thick) at an evaporation rate of 1.2 Å/s, which was patterned by an in-situ shadow mask to form an active emitting area of 4 mm^2^.

### 2.3. Characterizations

X-ray diffraction (XRD; D/Max-2200pc; Rigaku, Tokyo, Japan) was used to confirm the formation of the ZnMgO NP-based ETLs (comprising ZnO NPs, Zn_.9_Mg_0.1_O NPs, or ZnMgO NPs with an A/B core/shell structure), with Cu-Kα radiation applied to the centrifuged ZnMgO NPs. Field-emission transmission electron microscopy (FE-TEM) (Tecnai F30 S-Twin; JEOL Ltd., Tokyo, Japan) was used to determine the actual particle size of the ZnMgO-based ETLs. XPS measurements were conducted using a Nexsa XPS system (ThermoFisher Scientific, Waltham, MA, USA) and UPS analysis with a He (I) 21.22-eV gas discharge lamp, to characterize the O 1s level and valence band maximum (VBM) of the ZnMgO-based ETL thin films, respectively. The transmittance and reflectance were measured using a spectrophotometer (UV-1650PC; Shimadzu Corp., Kyoto, Japan), with normally incident monochromatic light at the sample surface side. Current density–voltage–luminance (*J*–*V*–*L*) was evaluated using a computer-controlled source meter (2400; Keithley Instruments, Cleveland, OH, USA) and luminance meter (LS100; Konica Minolta, Tokyo, Japan). Electroluminescence (EL) spectra were recorded using a spectroradiometer (CS1000; Konica Minolta).

## 3. Results and Discussion

Figure 1 shows the typical XRD patterns of the ZnO NPs, Zn_0.9_Mg_0.1_O NPs, ZnMgO core/shell A and ZnMgO core/shell B. All samples exhibited the hexagonal wurtzite structure of ZnO. The XRD patterns of ZnO NPs, Zn_0.9_Mg_0.1_O NPs, ZnMgO NP core/shell A and ZnMgO NP core/shell B showed reflection in the (100), (002), (101), (102), (110), (103) and (112) planes, which was attributed to the ZnO phase. Notably, the peaks broadened as the Mg content in the ZnO NPs increased (Figure 1). This is attributable to the ZnMgO NPs with the core/shell structure having a greater amount of Mg than Zn_0.9_Mg_0.1_O NPs. Notably, the Mg content in ZnMgO core/shell B was higher than that in ZnMgO core/shell A.

Figure 2 shows typical FE-TEM images and corresponding fast Fourier transform patterns for NPs with a single-crystalline configuration. Figure 2a,b show TEM images of ZnO NPs and Zn_0.9_Mg_0.1_O NPs, respectively. The average diameter of the ZnO NPs and Zn_0.9_Mg_0.1_O NPs was estimated to be 4.69 and 4.51 nm, respectively. Figure 2c,e show the average diameter of the core layers pre-synthesized for ZnMgO NP core/shells A and B (3.64 and 3.95 nm, respectively). After completing the shell formation process on the pre-synthesized core layer, the diameter of ZnMgO NP core/shells A and B was measured as 4.54 and 4.25 nm, respectively, as shown in Figure 2d,f. The average diameters of the completed ZnMgO NPs with the core/shell structure increased after the shell formation process. This implies that ZnMgO shell layers with higher Mg content were grown on the ZnMgO cores.

The thin films fabricated using the ZnO NPs, Zn_0.9_Mg_0.1_O NPs, ZnMgO NP core/shell A and ZnMgO NP core/shell B were further investigated by XPS. Figure 3 shows the O 1s peaks in the XPS spectra of the thin films, which carries information about the quality of the NPs. The thin films exhibited an asymmetric O 1s spectra. The spectra were deconvoluted into three binding states, attributed to oxygen in the metal oxide lattice (O-M) surrounded by Zn and Mg ions, oxygen vacancies O^2−^ ions (O-V) in oxygen-deficient regions and oxygen bonded in hydroxide (-OH) [32,33,34]. O-M, O-V and -OH in the thin films denote the areas of the components centered at 531.3, 532.7 and 533.3 eV, respectively, in the figure. O-T represents the O 1s peak as a whole. The O-M/O-T, O-V/O-T and -OH/O-T values were estimated (Figure 3). The ratios of the peak area (O_V_/O_T_) of ZnO NPs, Zn_0.9_Mg_0.1_O NPs, ZnMgO NP core/shell A and ZnMgO NP core/shell B were 7.45%, 9.97%, 11.88% and 23.13%, respectively. Thus, higher doping amounts of Mg increased the number of oxygen vacancies in the NPs.

The electronic energy level configuration of the ZnO NPs, Zn_0.9_Mg_0.1_O NPs, ZnMgO NP core/shell A and ZnMgO NP core/shell B was investigated using UPS. The UPS spectra coinciding with the secondary-electron cutoff and VBM regions are shown in Figure 4a,b. The VBM level was calculated from the incident photon energy of 21.22 eV, onset energy in the valence band region (E_onset_) and high-binding energy cutoff region (E_cutoff_) using the following equation: VBM = 21.22 − (E_cutoff_ − E_onset_) [35,36]. The VBM levels of the thin films of ZnO NPs, Zn_0.9_Mg_0.1_O NPs, ZnMgO NP core/shell A and ZnMgO NP core/shell B were estimated to be 6.94, 7.21, 7.66 and 7.87 eV below the vacuum level, respectively. Figure 4c plots (αhν)^2^ against the photon energy (hν), where α, h and ν are the absorption coefficient, Planck’s constant and radiation frequency, respectively. The absorption coefficient was calculated based on the UV/Vis data. The optical band gaps of the various films were evaluated using a Tauc plot of the UV/Vis absorption spectra. The optical band gaps of the films with the ZnO NPs, Zn_0.9_Mg_0.1_O NPs, ZnMgO NP core/shell A and ZnMgO NP core/shell B were estimated to be 3.39, 3.61, 3.76 and 3.70 eV below the vacuum level, respectively. Using the optical bandgap and VBM values, the CBM level of the ZnO NPs, Zn_0.9_Mg_0.1_O NPs, ZnMgO NP core/shell A and ZnMgO NP core/shell B were calculated to be 3.60, 3.62, 3.79 and 3.75 eV below the vacuum level, respectively. Figure 4d shows a schematic diagram of the flat-band energy levels of the QLEDs with ZnO NPs, Zn_0.9_Mg_0.1_O NPs, ZnMgO NP core/shell A and ZnMgO NP core/shell B. Figure 4e shows the schematic of the fabricated QLEDs with ZnMgO-based ETLs. Except for the vacuum evaporation of Al cathode, all layers were spin coated to prepare the QLEDs. Figure 4f shows a photographic image of a device. It is shown in Figure 4f that the device consisted of 4 active-light-emitting QLEDs. The device area was 25×25 mm^2^ and active-light-emitting area was 2×2 mm^2^.

Figure 5 shows the electroluminescent (EL) characteristics of the QLEDs with various ZnMgO NP-based ETLs. Figure 5a shows the current density (*J*) and luminance (*L*) curves as a function of the applied voltage (*V*). It is well known that the current density can be attributed not only to the electron/hole injection barrier, but also to the mobility of the ETL and HTL. Although the energy barrier between the Al cathode and ETL of ZnO NPs was larger compared to those between the Al cathode and ETLs of Zn_0.9_Mg_0.1_O NPs, ZnMgO NP core/shell A and ZnMgO NP core/shell B, the current density of the QLED with the ZnO NP ETL exhibited the highest value. This indicates that the mobility of the ZnO NP ETL should be the highest. Figure 5a also shows that the current density of QLED with Zn_0.9_Mg_0.1_O NPs was higher than those of QLEDs with a core/shell structure. This suggests that, for the ZnMgO NP shell layer with greater Mg content compared to the ZnMgO NP core layer, there was a reduction in the mobility of ZnMgO NPs due to an increase in oxygen vacancy content. From Figure 5a, the turn-on voltage of the QLEDs with ZnO NPs, Zn_0.9_Mg_0.1_O NPs, ZnMgO NP core/shell A and ZnMgO NP core/shell B was estimated to be 4.2, 6.1, 4.1 and 5.6 V, respectively; the turn-on voltage was defined as the voltage at which the luminance reached 1 cd/m^2^. The QLED with ZnMgO NP core/shell A exhibited the lowest turn-on voltage. Figure 5a also shows that the maximum luminance of the QLEDs fabricated with ETLs of ZnO NPs, Zn_0.9_Mg_0.1_O NPs, ZnMgO NP core/shell A and ZnMgO NP core/shell B (50,784.7, 88,235.0, 55,137.3 and 20,716.9 cd/m^2^, respectively). Figure 5b shows that the maximum current efficiency of the QLEDs with ETLs of ZnO NPs, Zn_0.9_Mg_0.1_O NPs, ZnMgO NP core/shell A and ZnMgO NP core/shell B was 21.2, 37.1, 58.0 and 45.4 cd/A, respectively. The maximum current efficiency of the QLEDs with Zn_0.9_Mg_0.1_O NPs, ZnMgO NP core/shell A and ZnMgO NP core/shell B was enhanced by 75%, 173.6% and 114.2%, respectively, compared to the QLED with a ZnO NP ETL. The maximum current efficiency of the QLEDs with core/shell ZnMgO NP A and core/shell ZnMgO NPs B was enhanced by 156.3% and 122.4%, respectively, compared to the QLED with a Zn_0.9_Mg_0.1_O NPs ETL. Because the ZnMgO NP shell layers, which have higher Mg content than the ZnMgO NP core layer, have more oxygen vacancies than Zn_0.9_Mg_0.1_O NPs without the shell, the mobility of electrons in ZnMgO NP ETLs with a core/shell structure should be lower than that of a ZnMgO NP ETL without a core/shell structure. Thus, the optimal charge balance was achieved in the QLED with a ZnMgO NP core/shell A ETL, likely due to appropriate reduction of the electron mobility and energy barrier between the cathode and ETL with ZnMgO core/shell layers, which enhanced the charge balance. Figure 5c shows that the maximum power efficiency of the QLEDs with ZnO NP ETL, Zn_0.9_Mg_0.1_O NP ETL, ZnMgO NP core/shell A ETL and ZnMgO NP core/shell B ETL was 10.9, 10.9, 23.3 and 9.6 lm/W, respectively. The maximum power efficiency of the QLED with ZnMgO NP core/shell A ETL was enhanced by 113.8%, compared to the QLED with ZnO NP ETL and Zn_0.9_Mg_0.1_O NP ETL. It is obvious that the QLED with ZnMgO NP core/shell A ETL considerably enhances the power efficiency in comparison with the QLEDs with ZnO NP and Zn_0.9_Mg_0.1_O NP ETLs without core/shell structure. Figure 5d shows the PL spectrum from the CdSe/ZnS QD and the EL spectra from the QLEDs with ZnO NP ETL, Zn_0.9_Mg_0.1_O NP ETL, ZnMgO NP core/shell A ETL and ZnMgO NP core/shell B ETL. The characteristic parameters of the spectra are summarized in Table 1. There is no parasite blue emission from PVK so that the color purity of the device is ensured. This also indicates that charge recombination occurred only in the EML. We note that there are blue-shifted EL peaks for QLEDs with ZnO NPs, Zn_0.9_Mg_0.1_O NPs, ZnMgO NP core/shell A and ZnMgO NP core/shell B compared to the PL peak of the QD. We attribute this blue-shift to the electric-field induced Stark effect [37,38] under high volage and current.

## 4. Conclusions

ZnO-based metal oxide NPs of ZnO, Zn_0.9_Mg_0.1_O, ZnMgO NP core/shell A and ZnMgO NP core/shell B were synthesized as an ETL for QLEDs and highly efficient QLEDs were developed with the NPs. XRD and TEM analyses revealed the single-crystalline nature of the NPs. The electronic energy level structure and oxygen vacancy content of the ZnO NPs, Zn_0.9_Mg_0.1_O NPs, ZnMgO NP core/shell A and ZnMgO NP core/shell B were investigated using UPS and XPS, respectively. The maximum current efficiency and maximum power efficiency of the QLEDs with a ZnMgO NP core/shell A ETL were enhanced by 156.3% and 113.8%, respectively, compared to the QLED with a Zn_0.9_Mg_0.1_O NP ETL. As a result, an appropriate balance of the mobility of the ETLs and energy barrier between the cathode and ETL was achieved in the QLED with ZnMgO NP core/shell A; the improvement in current efficiency and power efficiency is likely due to appropriate reduction of the electron mobility and energy barrier between the Al cathode and ZnMgO NP ETL with a core/shell structure.

## Figures and Tables

**Figure 1 materials-16-00600-f001:**
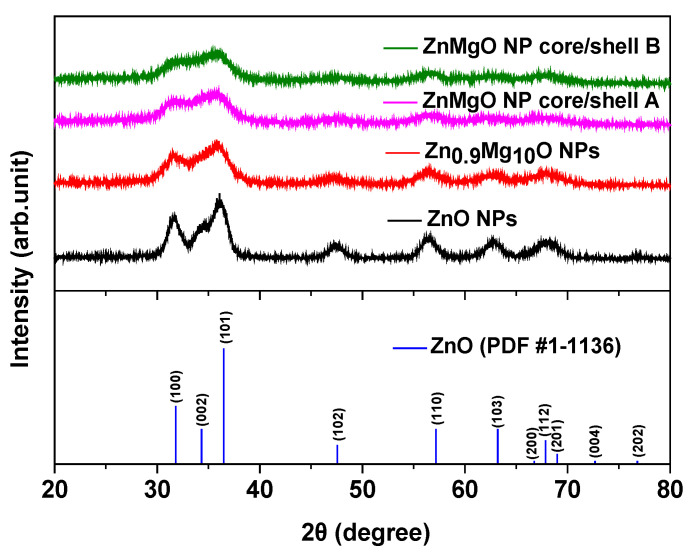
X-ray diffraction patterns of the ZnO nanoparticles (NPs), Zn_0.9_Mg_0.1_O NPs, ZnMgO NP core/shell A, ZnMgO NP core/shell B and bulk wurtzite ZnO over a *2θ* scanning range of 20–80°.

**Figure 2 materials-16-00600-f002:**
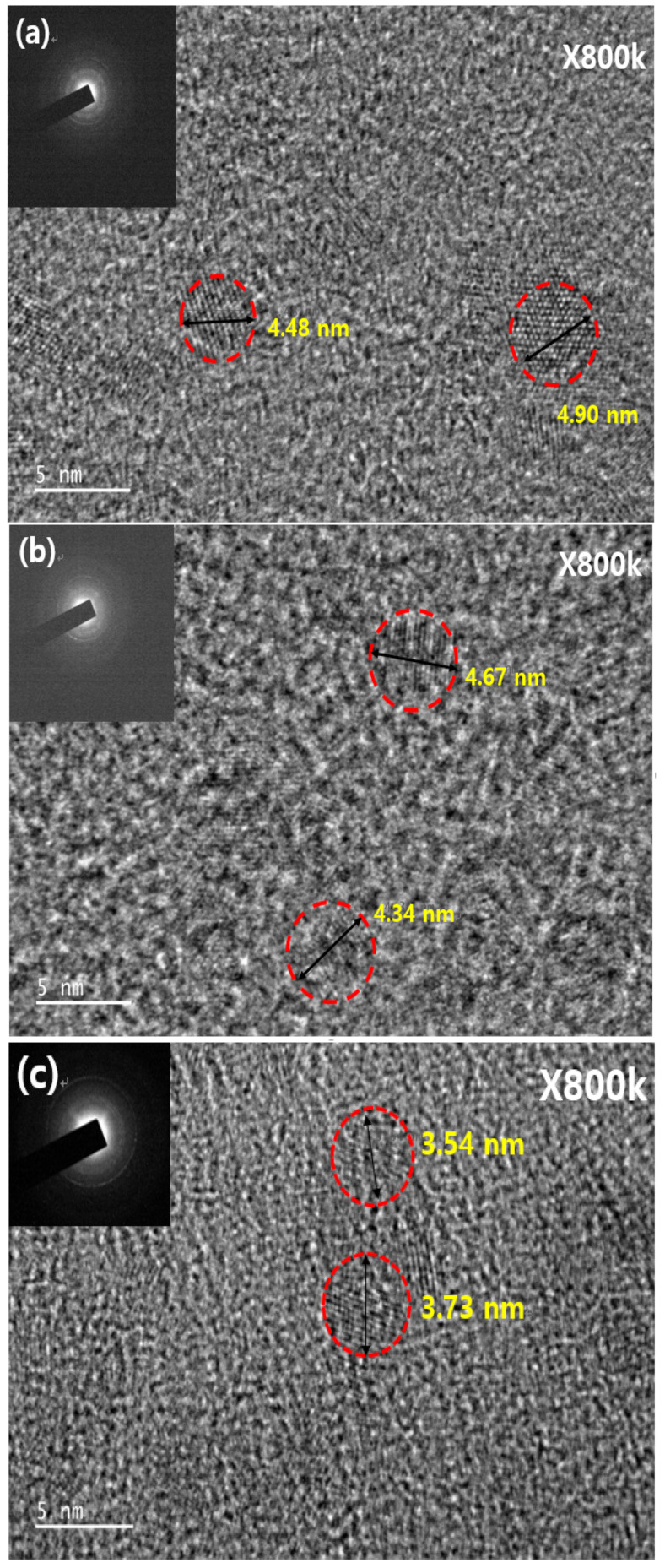
Field-emission transmission electron microscopy images of the (**a**) ZnO NPs, (**b**) Zn_0.9_Mg_0.1_O NPs, (**c**) core layer pre-synthesized for the ZnMgO NP core/shell A, (**d**) ZnMgO NP core/shell A, (**e**) core layer pre-synthesized for the ZnMgO NP core/shell B and (**f**) ZnMgO NP core/shell B. The insets illustrate the FFT patterns of the corresponding samples.

**Figure 3 materials-16-00600-f003:**
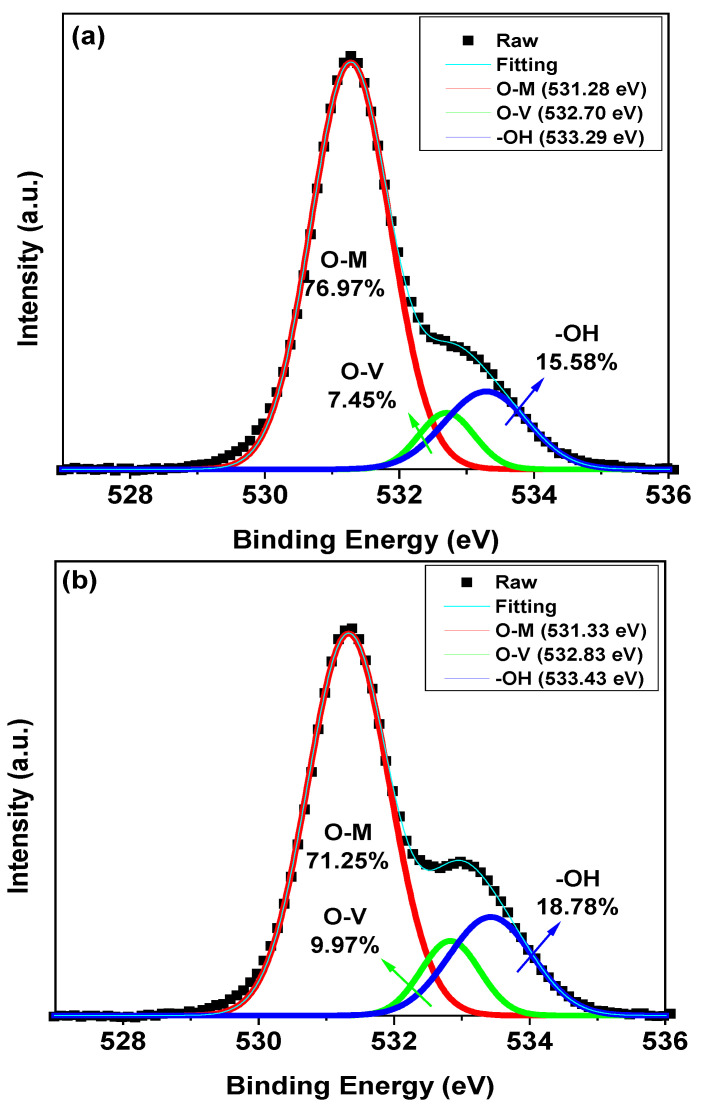
X-ray photoelectron spectroscopy spectra of (**a**) ZnO NPs, (**b**) Zn_0.9_Mg_0.1_O NPs, (**c**) ZnMgO NP core/shell A and (**d**) ZnMgO core/shell B.

**Figure 4 materials-16-00600-f004:**
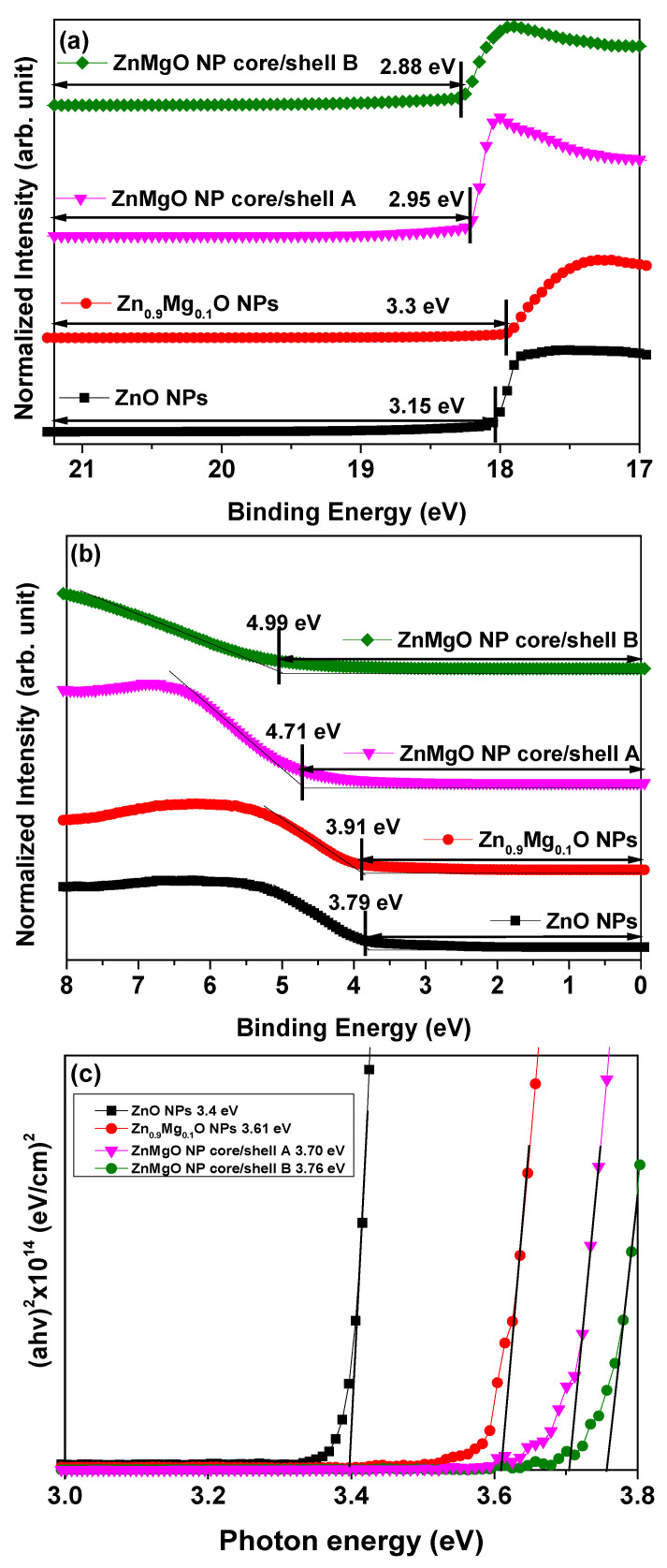
Ultraviolet photoelectron spectroscopy (UPS) spectra of the ZnO NPs, Zn_0.9_Mg_0.1_O NPs, ZnMgO NP core/shell A and ZnMgO NP core/shell B: (**a**) secondary electron cutoff and (**b**) valence band maximum regions from the UPS spectra. (**c**) (αhν)^2^ plots against the photon energy (hν) converted from the absorption spectra. (**d**) Schematic energy band diagram of the quantum dot light-emitting diodes (QLEDs). (**e**) Schematic diagram of the QLED configuration (**f**) Photographic image of device showing four active-light-emitting area.

**Figure 5 materials-16-00600-f005:**
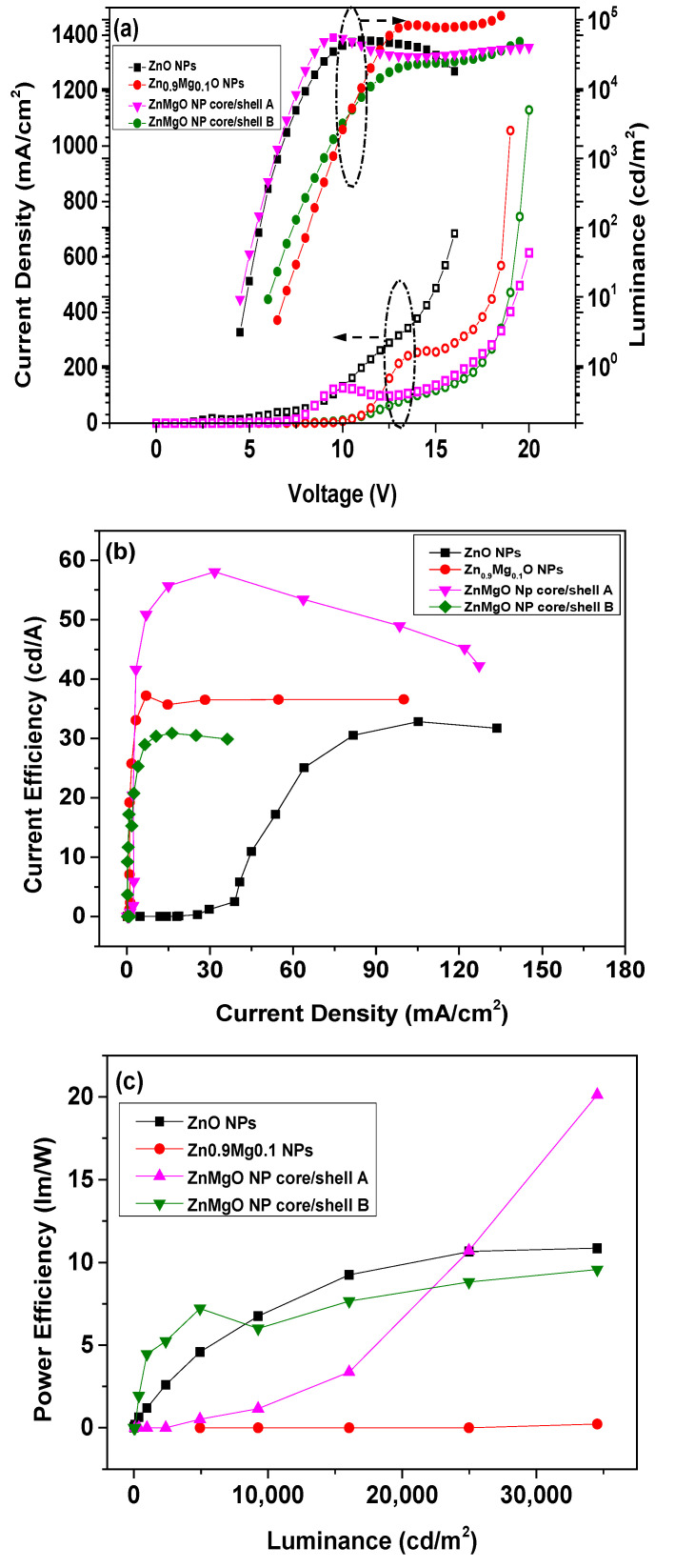
Electroluminescence performance of the QLEDs with electron transport layers (ETLs) prepared from ZnO NPs, Zn_0.9_Mg_0.1_O NPs, ZnMgO NP core/shell A and ZnMgO NP core/shell B. (**a**) Current density and luminance curves as a function of the applied voltage. (**b**) Current efficiency curves as a function of the current density. (**c**) Power efficiency curves as a function of the luminance. (**d**) The PL spectrum of the QD and the EL spectra of the QLEDs with ZnO NP ETL, Zn_0.9_Mg_0.1_O NP ETL, ZnMgO NP core/shell A ETL and ZnMgO NP core/shell B ETL.

**Table 1 materials-16-00600-t001:** The key parameters of the spectra, including the PL of the QD and the EL of QLEDs with ETLs of ZnO NPs, ZnMgO NPs, Zn_0.9_Mg_0.1_O NP core/shell A and ZnMgO NP core/shell B.

Sample	Peak (nm)	FWHM (nm)
PL of QD	542.4	33.3
EL of QLED with ZnO NPs	536.1	40.6
EL of QLED with Zn_0.9_Mg_0.1_O NPs	535.0	40.9
EL of QLED with ZnMgO NP core/shell A	527.0	37.3
EL of QLED with ZnMgO NP core/shell B	527.8	37.4

## Data Availability

The data that support the findings of this study are available from the corresponding author upon reasonable request.

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
