# Peer review of "Performance Improvement of Quantum Dot Light-Emitting Diodes Using a ZnMgO Electron Transport Layer with a Core/Shell Structure"

_materials, 2023, doi:10.3390/ma16020600_

Round 1
Reviewer 1 Report
The authors reported that a ZnMgO NP ETL with a core/shell structure developed to create highly efficient QLEDs. The QLED incorporating the as-fabricated ZnMgO NP ETL with core/shell structure exhibited a maximum luminance of 55,137.3 cd/m2 and maximum current efficiency of 58.0 cd/A. The maximum current efficiency of the QLEDs with ZnMgO NP core/shell A improved by as much as 156.3% compared to the QLED with a Zn0.9Mg0.1O NP ETL, thus demonstrating the benefits of ZnMgO NPs with the specified core/shell structure. The manuscript is clearly expressed and the experimental results can well support the claim. The manuscript can be accepted after revision as follows:
1. Some references (Journal of Colloid and Interface Science 529 (2018) 205-213; Molecules 26 (2021)3922) should be cited.
2. The scale bars in Figure 3(a)-(d) are not clear, it is recommended to process it again.
3. There are some spelling, grammatical, and other errors in the manuscript, which should be carefully checked.
Author Response
Thank you very much for reviewers’ comments which will definitely help make the paper a more valuable reference. According to the comments of the reviewer, the manuscript has been thoroughly revised. I hope that the revisions based on the comments meet the requirements, and I look forward to a positive response.
We appreciate your time in reviewing our manuscript.
Thank you,
Chang Kyo Kim
1. Some references (Journal of Colloid and Interface Science 529 (2018) 205-213; Molecules 26 (2021)3922) should be cited.
Ans. 1)
he references recommended by Reviewer are referred in Introduction. Thank you for your recommendation of the references.
2. The scale bars in Figure 3(a)-(d) are not clear, it is recommended to process it again.
Ans. 2) We thank reviewer’s kind advice. We processed it again as you recommended.
3. There are some spelling, grammatical, and other errors in the manuscript, which should be carefully checked.
Ans. 3)
We thoroughly checked and revised English again. The English in this document has been checked by at least two professional editors, both native speakers of English. For a certificate, please see:
http://www.textcheck.com/certificate/gTCA8v

Reviewer 2 Report
The paper “Performance Improvement of Quantum Dot Light-Emitting Diodes Using a ZnMgO Electron Transport Layer with a Core/Shell Structure” shows improvement on QLED parameters by layer composition modification but needs major revision to clearly explain experiments and results before publication. Moreover, it lacks electroluminescence spectra of the devices to see for what applications it would be suitable.
Production methods have to be clarified:
It should be described how ITO was patterned,
PVK solution concentration and PVK spincoating parameters have to be provided,
CdSe/ZnS QDs spincoating parameters have to be provided (spincoated, rpm, acceleration, duration, temperature),
ZnMgO-based ETLs comprising ZnO NPs, ZnMgO NPs, and ZnMgO NPs with a core/shell structure (A/B) spincoating parameters have to be provided,
Al cathode deposition method parameters and used equipment have to be provided.
In Fig. 1 PDF card and numbered ZnO peaks have to be provided for correlation with experiment.
EL spectra of the device are not shown. They have to be provided at different current and properly commented and compared to literature data. Device photo has to be provided, the dimensions of device have to be indicated. Scattering of efficiency statistics on the devices have to be calculated to verify the improvement is not only scattering of device parameters.
Power efficiency of the LEDs have to be provided and compared to available data.
The LED diagram in Fig. 4 a has to be provided for each diode structure, now it is not understandable. EML is not defined in the abstract, EML acronym does not correlate with light emitting layer.
Provided percent compositions are provided with 0.01 percent precision, it is doubtable, errors have to be evaluated.
Author Response
Thank you very much for reviewers’ comments which will definitely help make the paper a more valuable reference. According to the comments of the reviewer, the manuscript has been thoroughly revised. I hope that the revisions based on the comments meet the requirements, and I look forward to a positive response.
We appreciate your time in reviewing our manuscript.
Thank you,
Chang Kyo Kim
Production methods have to be clarified:
Thank you for valuable comments about production method. We revised them as you commented.
1. It should be described how ITO was patterned,
Ans. 1) We added the method the ITO was patterned.
2. PVK solution concentration and PVK spin coating parameters have to be provided,
Ans. 2) We added the PVK solution concentration and PVK spin coating parameters as you suggested.
3. CdSe/ZnS QDs spin coating parameters have to be provided (spincoated, rpm, acceleration, duration, temperature),
Ans. 3) We added the QD spin-coating parameters such as rpm, acceleration, duration, and temperature.
4. ZnMgO-based ETLs comprising ZnO NPs, ZnMgO NPs, and ZnMgO NPs with a core/shell structure (A/B) spincoating parameters have to be provided,
Ans. 4) We described the spin coating parameters of ZnMgO-based ETLs comprising ZnO NPs, ZnMgO NPs, and ZnMgO NPs with a core/shell structure (A/B).
5. Al cathode deposition method parameters and used equipment have to be provided.
Ans. 5) We described Al cathode deposition method parameters and used equipment.
6. In Fig. 1 PDF card and numbered ZnO peaks have to be provided for correlation with experiment.
Ans. 6) Thank you for your valuable comment. We revised it in Figure 1.
7. EL spectra of the device are not shown. They have to be provided at different current and properly commented and compared to literature data.
Ans. 7) Thank you for good question. We provided the EL spectra of the devices in Figure 5(d). The PL spectrum of the QD was also provided to compare with the EL spectra of the devices.
8. Device photo has to be provided, the dimensions of device have to be indicated.
Ans. 8. Thank you for your comment. We provided device photo with the dimension in Figure 4(f).
9. Scattering of efficiency statistics on the devices have to be calculated to verify the improvement is not only scattering of device parameters.
Ans. 9) Thank you for your good comment. We fabricated 8 of the same devices, measured the device characteristics, and checked the average performance. We presented the average performance data so that the presented data is not scattering of device parameters.
10. Power efficiency of the LEDs have to be provided and compared to available data.
Ans. 10) Thank you for your kind comment. Power efficiency of the LEDs is provided in the Figure 5(c) of the revised manuscript.
11. The LED diagram in Fig. 4 a has to be provided for each diode structure, now it is not understandable.
Ans. 11) Thank you for your good comment. We revised a band diagram showing 4 different ZnMgO-based ETLs in Figure 4(d) to compare their band structures at a glance.
12. EML is not defined in the abstract, EML acronym does not correlate with light emitting layer.
Ans12) Thank you for your comment. We revised the EML into emitting layer in the abstract. We also revised light emitting layer (EML) into emitting layer (EML) to correlate the acronym of EML in Introduction section.
13. Provided percent compositions are provided with 0.01 percent precision, it is doubtable, errors have to be evaluated.
Ans. 14) Thank you for your good comment. In order to synthesize a solution with precise compositions, we weighed the precursor materials using an electronic scale with a precision degree of 0.0001 g.

Reviewer 3 Report
Dear Editor,
Thanks for inviting me for reviewing the manuscript entitled Performance Improvement of Quantum Dot Light-Emitting Diodes Using a ZnMgO Electron Transport Layer with a Core/Shell Structure authored by Y. B. Eun and C. K. Kim, et al.
The author studied the ZnMgO NP ETL with a core/shell structure that exhibited a maximum luminance of 55,137.3 cd/m2 and maximum current efficiency of 58.0 cd/A. There are many problems throughout the manuscript, and the revised version can be resubmitted after they are fully solved.
1) First of all, the language should be checked throughout the manuscript, there are some grammar errors and unclear expression.
2) The abstract is not written well and should be re-organized.
3) The author claimed the core/shell structure of ZnMgO however the core and shell formation are only assumed by synthetic conditions and size analysis from TEM images. A real shell with different crystal lattice or mismatched lattice differential from core is not identified. Thus higher resolution TEM image should be provided.
Since the importance of the size statistic of ZnMgO or claimed core/shell structures, the number of those quantum dots should be large enough. The authors are suggested to provide better TEM images.
4) The title is on quantum dot LED, while the material used is ZnMgO nanoparticles, would the term of ZnMgO quantum dots better?
Author Response
Thank you very much for reviewers’ comments which will definitely help make the paper a more valuable reference. According to the comments of the reviewer, the manuscript has been thoroughly revised. I hope that the revisions based on the comments meet the requirements, and I look forward to a positive response.
We appreciate your time in reviewing our manuscript.
Thank you,
Chang Kyo Kim
1. First of all, the language should be checked throughout the manuscript, there are some grammar errors and unclear expression.
Ans1) Thank you for your comment. We thoroughly checked English and revised the manuscript. The English in this document has been checked by at least two professional editors, both native speakers of English. For a certificate, please see:
http://www.textcheck.com/certificate/gTCA8v
2. The abstract is not written well and should be re-organized.
Ans. 2) Thank you for your comment. We re-organized the abstract.
3. The author claimed the core/shell structure of ZnMgO however the core and shell formation are only assumed by synthetic conditions and size analysis from TEM images. A real shell with different crystal lattice or mismatched lattice differential from core is not identified. Thus higher resolution TEM image should be provided. Since the importance of the size statistic of ZnMgO or claimed core/shell structures, the number of those quantum dots should be large enough. The authors are suggested to provide better TEM images.
Ans. 3) Thank you for your good comment. We are sorry that we cannot provide the TEM images with higher resolution. It is because that we cannot take TEM images at this time. The core and shell are composed of the same ZnMgO as homostructure. Mg content in the shell layer is high compared with that of the core. Since the size of an Mg atom is very close to that of a Zn atom, the structural mismatch at the interface between the shell and the core is not expected to be large. This can be explained from the fact that there is no significant shift in diffraction position with increasing Mg content into ZnO NPs from XRD analysis. It indicates that very little distortion is caused by doping Mg into ZnO NPs.
4. The title is on quantum dot LED, while the material used is ZnMgO nanoparticles, would the term of ZnMgO quantum dots better?
Ans. 4) Thank you for your kind advice. In General, light-emitting layer is used as a name of display device. In this study, CdSe/ZnS quantum dot was used as an emitting layer. Since the ZnMgO NPs with core/shell structure employed in this study were used as an electron transport layer (ETL) in the QLEDs, we don’t think that it is good to use ZnMgO quantum dot in the title.
